

# Human-wildlife conflicts with crocodilians, cetaceans and otters in the tropics and subtropics

Patrick Cook[1,2], Joseph E. Hawes[3,4,5], João Vitor Campos-Silva[4,5,6,7] and Carlos A. Peres[1,5]

[1] School of Environmental Sciences, University of East Anglia, Norwich, United Kingdom
[2] Biological and Environmental Sciences, University of Stirling, Stirling, United Kingdom
[3] Applied Ecology Research Group, School of Life Sciences, Anglia Ruskin University, Cambridge, United Kingdom
[4] Faculty of Environmental Sciences and Natural Resource Management (MINA), Norwegian University of Life Sciences, Ås, Norway
[5] Instituto Juruá, Manaus, Amazonas, Brazil
[6] Instituto de Ciências Biológicas e da Saúde, Universidade Federal de Alagoas, Maceió, Alagoas, Brazil
[7] Instituto Nacional de Pesquisas da Amazônia, Manaus, Amazonas, Brazil

Corresponding author
Patrick Cook, patrick.cook@stir.ac.uk

## ABSTRACT

Conservation of freshwater biodiversity and management of human-wildlife conflicts are major conservation challenges globally. Human-wildlife conflict occurs due to attacks on people, depredation of fisheries, damage to fishing equipment and entanglement in nets. Here we review the current literature on conflicts with tropical and subtropical crocodilians, cetaceans and otters in freshwater and brackish habitats. We also present a new multispecies case study of conflicts with four freshwater predators in the Western Amazon: black caiman (*Melanosuchus niger*), giant otter (*Pteronura brasiliensis*), boto (*Inia geoffrensis*) and tucuxi (*Sotalia fluviatilis*). Documented conflicts occur with 34 crocodilian, cetacean and otter species. Of the species reviewed in this study, 37.5% had conflicts frequently documented in the literature, with the saltwater crocodile (*Crocodylus porosus*) the most studied species. We found conflict severity had a positive relationship with species body mass, and a negative relationship with IUCN Red List status. In the Amazonian case study, we found that the black caiman was ranked as the greatest 'problem' followed by the boto, giant otter and tucuxi. There was a significant difference between the responses of local fishers when each of the four species were found entangled in nets. We make recommendations for future research, based on the findings of the review and Amazon case study, including the need to standardise data collection.

## INTRODUCTION

Conflict between humans and wildlife poses a major challenge for biological conservation (*Dickman, 2010*). Human-wildlife conflicts arise as a result of recurring negative interactions between humans and wildlife and are frequently deep rooted in social beliefs (*Pimm & Raven, 2000*). Understanding the underlying factors driving conflicts is integral to successful management, due to the often-increasing proximity between humans and wildlife, driven by growing human populations and the recovery of rare, conflict-generating species (*Inskip & Zimmermann, 2009*; *Groenendijk et al., 2014*). Increasing our knowledge of conflicts in freshwater and brackish ecosystems, between humans and piscivores, is especially important in the tropics and subtropics, given the heavy exploitation pressure and continued decline of wildlife populations (*He et al., 2019*).

Freshwater habitats cover approximately 3% of the Earth's land surface area (*Pekel et al., 2016*), exposing vertebrates to potential conflicts with disproportionately high densities of humans, as a result of overlapping distributions and utilisation of similar resources (*Woodroffe & Ginsberg, 1998*; *Treves & Karanth, 2003*; *Dudgeon et al., 2006*). Piscivores can impose significant impacts on human livelihoods in freshwater and brackish environments, including attacks on people and damage to fishing gear, in addition to co-depletion of fish stocks (*Rosas-Ribeiro, Rosas & Zuanon, 2012*; *Sideleau & Britton, 2013*). Managing such conflicts to ensure long-term persistence of wildlife populations is vital to maintaining ecosystem integrity (*Rio et al., 2001*). Yet this is particularly challenging in tropical and subtropical regions where freshwater fisheries more often represent a critical component of the subsistence diets and commercial revenues of local people (*Michalski et al., 2012*).

Potential conflicts within marine fisheries have been well documented and show negative impacts on both the conservation of large marine predators and the socio-economic viability of fishing activities (*Tixier et al., 2021*). However, despite the importance of inland fisheries, a review of the conflicts reported in freshwater and inland brackish systems such as estuaries and lagoons has not yet been undertaken. Here, we address this research gap by investigating conflicts between humans and three major groups of piscivores (crocodilians, cetaceans and otters) found throughout the tropics and subtropics, through (i) a case study from western Brazilian Amazonia, and (ii) a global literature review.

Crocodilians, cetaceans and otters are responsible for high levels of conflict in the Amazon (*Loch, Marmontel & Simões-Lopes, 2009*; *Fonseca & Marmontel, 2011*; *Alves, Zappes & Andriolo, 2012*; *Lima, Marmontel & Bernard, 2014a*), one of the most important freshwater habitats on Earth where the majority of rural people depend on fisheries (*Begossi et al., 2018*). Yet most human-wildlife conflicts in the vast Amazonian basin are likely to be unreported; our case study surveys a remote region with no previous documentation of human-wildlife conflicts. We specifically tested: (1) which species are most involved in human-wildlife conflicts; and (2) what types of conflicts occur for each species. Distance to the nearest urban area or access to aquatic habitats (such as seasonally flooded forests) could influence conflict severity by altering the probability of

interactions occurring between humans and wildlife (*Rosas-Ribeiro, Rosas & Zuanon, 2012*). We therefore also tested (3) whether proximity to the nearest town predicts human-wildlife conflict severity; and (4) whether percentage of seasonally flooded forest around communities predicts human-wildlife conflict severity.

To provide a global context for this case study, we conducted a quantitative literature review, which we restricted to the same three taxonomic groups for consistency. Specific research questions for the literature review include: (1) How many and which species of crocodilians, cetaceans and otters are involved in human-wildlife conflicts? (2) What types of human-wildlife conflicts occur? (3) What is the frequency of conflict documented in the primary literature for each species? (4) What is the conflict severity for each species?

Species involved in conflicts are often large-bodied and slow to reproduce, and their population status can be directly or indirectly affected by the conflict (*Alves, Zappes & Andriolo, 2012*; *Huang et al., 2012*; *Groenendijk et al., 2014*). Conflict severity may be expected to be positively related to body mass, due to the greater threat posed to human life by larger-bodied animals, as well as increased potential damage to fishing equipment and greater risk of the species being exploited. It may also be expected that conflict severity would be higher for less threatened species (as indicated by IUCN Red List status), due to more frequent human interactions with more common species. As additional research questions, we therefore tested: (5) how well body mass predicts human-wildlife conflict severity; and (6) how well IUCN threat status predicts human-wildlife conflict severity.

## MATERIALS AND METHODS

### Focal study area

Our focal landscape study was conducted in the state of Amazonas, Brazil along the mid-section of the Juruá River in two contiguous sustainable-use forest reserves: the Médio Juruá Extractive reserve (ResEx Médio Juruá), and the Uacari Sustainable Development Reserve (RDS Uacari) (Fig. 1). These two reserves are home to a combined total of approximately 4,000 rural Amazonians, living in 58 communities and employed in a diverse range of extractive livelihoods (*Newton, Endo & Peres, 2012*). Communities typically have access to extensive floodplains and are located along the main river channel or on oxbow lakes, which are embedded within forests that are seasonally flooded by nutrient-rich white-water, known as *várzea* (*Hawes et al., 2012*). Communities are therefore deeply entwined with their aquatic environment, and fishing represents both the principal source of protein in the subsistence diet of reserve residents (*Endo, Peres & Haugaasen, 2016*), and one of the main sources of disposable income (*Batista et al., 1998*). Our focal study reserves represent an important site for globally significant community-based conservation arrangements (*Campos-Silva & Peres, 2016*; *Campos-Silva et al., 2018*) that benefit a wide range of large-bodied freshwater piscivores, including the black caiman (*Melanosuchus niger*), the giant otter (*Pteronura brasiliensis*), and two cetaceans: the Amazon river dolphin or boto (*Inia geoffrensis*) and the tucuxi (*Sotalia fluviatilis*) (Fig. 2).

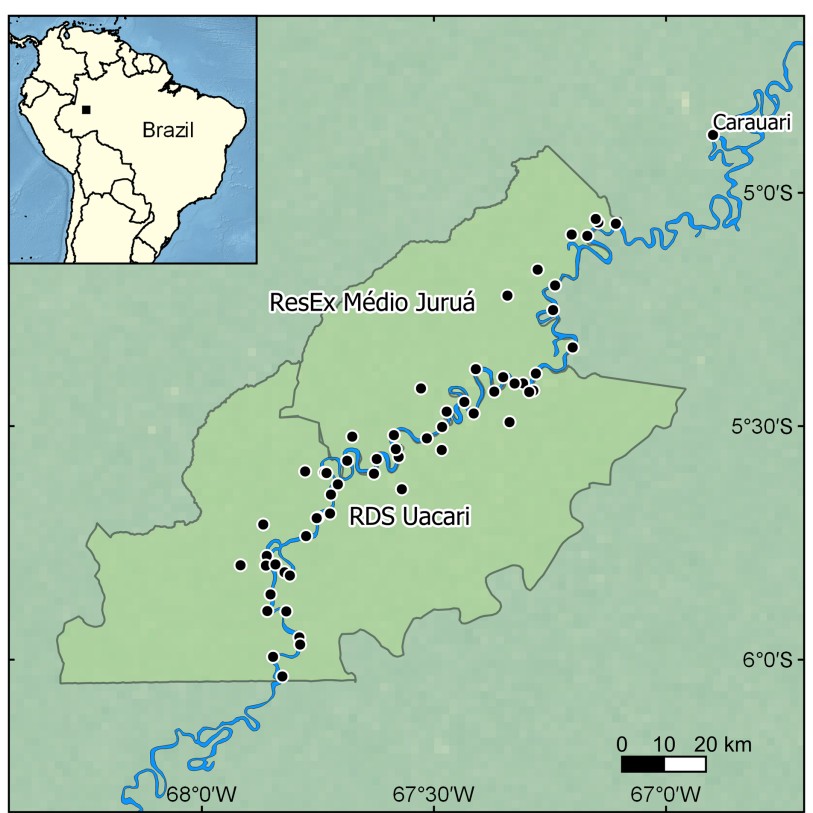

**Figure 1 Map showing the location of the focal study landscape in the Médio Juruá region of western Brazilian Amazonia.** Interviews were conducted in local communities (black dots) along the Juruá River (blue line) within two sustainable-use reserves (light green polygons). Made with Natural Earth. Free vector and raster map data at https://www.naturalearthdata.com/.

## Fisher interviews

We employed a semi-structured questionnaire design to investigate perceptions of human-wildlife conflict involving four species of aquatic fauna: black caiman, giant otter, boto and tucuxi. Two interviewers (JEH & JVCS) conducted a total of 49 interviews at 37 local communities located within the two sustainable-use reserves in the Médio Juruá region (Fig. 1), during September–November 2014. We selected interviewees non-randomly, targeting the most experienced fishers in each community (either one individual or a small group of individuals). Interviews typically lasted 30 min and interviewees rank-ordered the potential problem of the conflict caused, with 1 being the greatest problem and 4 the least (*Michalski et al., 2012*). In addition, we included eight objective yes/no questions asking whether any of the focal species cause problems, damage equipment, become entangled in nets, frighten away fish, or cause the interviewee to leave an area to fish elsewhere when the species has been sighted, and whether co-existence with the focal species in the future depends on continued increases in its population. Interviewees were asked what their most likely response would be to finding one of the four species entangled in their fishing nets, such as killing or releasing the individual, if the species could escape of its own accord or if it was likely to die before being found.

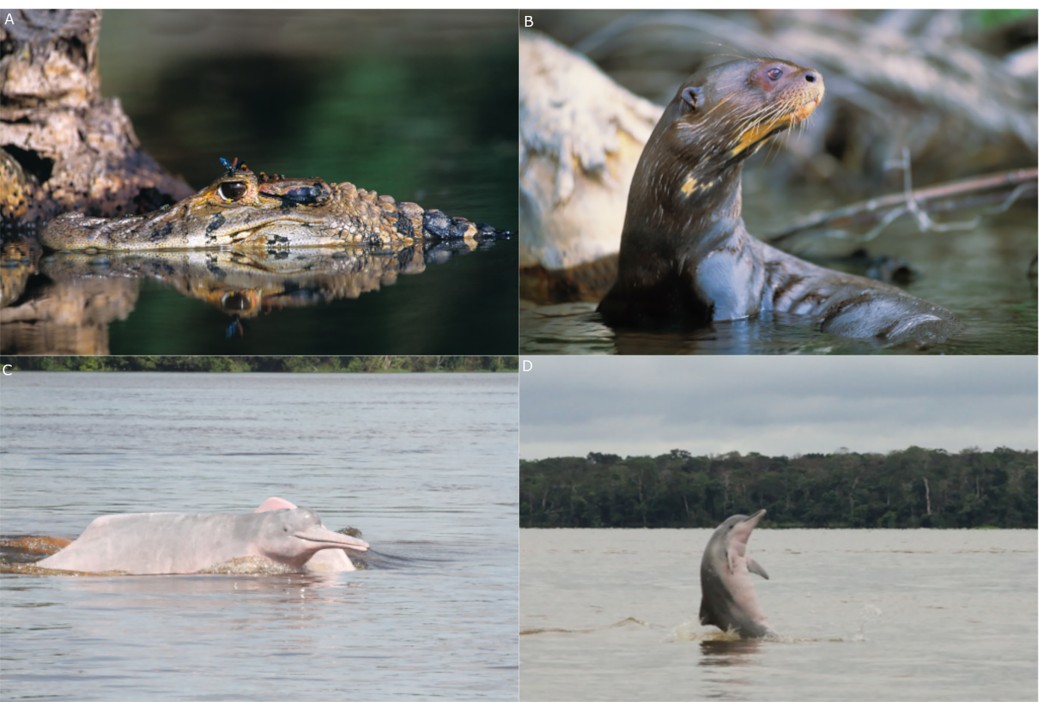

**Figure 2** **The four 'problem species' in the mid-Juruá.** (A) Black caiman, (B) giant otter, (C) boto and (D) tucuxi. Photo credits: (A) and (B) Frank Hajek and Jessica Groenendijk; (C) and (D) Sannie Brum.

Interviewees were also asked if the species had been hunted in the community, hunted within the last 2 years (2013 or 2014), or hunted within the informant's lifetime.

We thank the Secretaria do Estado do Meio Ambiente e Desenvolvimento Sustentável (SDS-DEMUC) and the Instituto Chico Mendes de Conservação da Biodiversidade (ICMBio) for authorising this research (permit number: 45054-1).

## Literature review

To provide a global context for our local case study, we conducted a literature review of human-wildlife conflicts, for all species of tropical and subtropical freshwater crocodilians, cetaceans and otters. Piscivores other than crocodilians, cetaceans and otters were not included in the study to ease comparison between the local case study and the wider literature review. Studies conducted outside the tropics and subtropics (35 degrees north and south of the equator) were excluded to ensure that data gathered from the literature review and case study were within the same latitudinal range. We focused on freshwater and brackish habitats, excluding any studies only from marine habitats. Similarly, studies that focused on interactions with fisheries were restricted to wild fisheries and excluded human-made fisheries such as aquaculture. We also excluded reports of attacks by animals in captivity, and attacks by wild animals on either livestock or pets.

Primary literature sources were collated from Google Scholar and Scopus. Two researchers (PC & JEH) conducted a general search using the keywords: conflicts, crocodilians, cetaceans and otters, before using a Boolean search string search that
**Table 1 Criteria for categories (adapted from _Inskip & Zimmermann, 2009_) used to determine the severity of conflicts found in the literature review, and the frequency of studies reporting conflicts.**

| Category | Definition |
| --- | --- |
| Severity of conflict | |
| Severe | Very high number of documented attacks on people (>20 reported fatalities and/or >50 non-fatal attacks[*]) |
| High | High number of documented attacks on people (1–20 fatalities and/or 10–50 non-fatal attacks[*]) |
| Moderate | Low number of documented attacks on people (1–9 non-fatal attacks[*]) |
| Low | No documented attacks on people[*] Other forms of conflict documented in the literature but not in relation to attacks on people[**] |
| Data Deficient | No documented attacks on people or evidence of other conflicts |
| Frequency of studies | |
| Frequent | Conflict documented in 5 or more primary literature sources |
| Infrequent | Conflict documented in 2–4 primary literature sources |
| Rare | Conflict documented in 1 primary literature source |
| Data Deficient | Conflict not documented in any primary literature sources |

Notes:
[*] We defined the number of fatal and non-fatal attacks on humans worldwide by crocodilians between 2008 and 2013 following _Sideleau & Britton (2013)_; it was not possible to filter the locations of these attacks to just the tropics or subtropics. We defined the number of non-fatal attacks by otters between 2000 and 2009 following _Belanger et al. (2011)_, filtering the data to the tropics and subtropics. The year of attacks were not provided in either of these studies, and we therefore use the different time periods given for crocodilians and otters.
[**] Other forms of conflicts may include but are not exclusive to: depredation of fish, damage to fishing gear and entanglement in fishing gear. It was not possible to quantify these types of conflicts, so we used their documentation in the literature as a substitute. This was only carried out for species with no documented attacks on humans.

included the common or scientific name of a species, together with the following keywords: attack, conflict, depredation, entanglement, perceptions and damage. The string search was conducted in the following format (("common name" OR "synonym" OR "scientific name" ) AND ( attack* OR conflict OR depred* OR entangle* OR perce* OR damage )). Additional articles were located through searching reference lists (snowballing) and subsequent citations (reverse snowballing). All keyword searches were conducted in English, which may have excluded some studies. For each literature source we documented the country of study and categorised the broad types of conflict documented such as attacks, net damage, depredation, entanglement and perceptions.

For each species, we categorised the frequency of documented conflict based on the number of primary literature sources referring to conflict with the species, 5 or more being frequently documented, 2–4 being infrequently documented and 1 being rarely documented (Table 1). We described the severity of conflict for each species, based on criteria adapted from _Inskip & Zimmermann (2009)_ (Table 1). Body mass in kilograms was attributed using the following categories: ≤10 kg, 11–49 kg, ≥50 kg adapted from _Inskip & Zimmermann (2009)_ and using data from the literature (_Macdonald, 2009_; _Hunter, 2011_; _Lakin et al., 2020_). We also assigned Red List Status using the IUCN Red List (_IUCN, 2020_). Category scores were assigned by PC and independently scored by JEH to check for inter-observer consistency.

## Data analysis

In the case study, we used a chi-squared test to determine if the species differed in their ranking as a 'problem species'. A Mann–Whitney _U_ test was used to investigate the level of

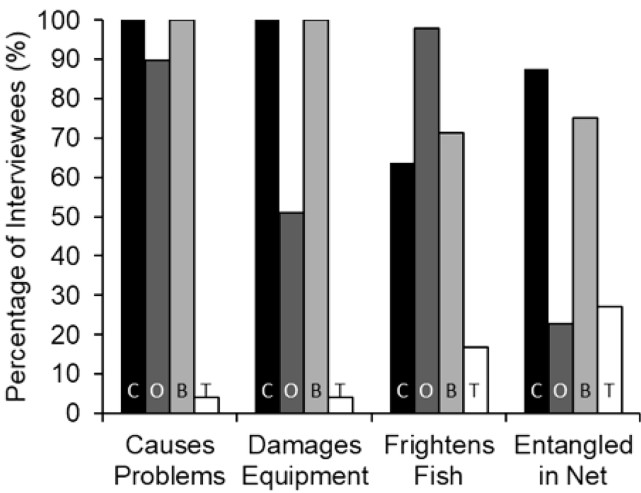

**Figure 3** **Percent of interviewees (*n* = 49) who indicated that each of the four species causes problems in general, damage fishing equipment, frightens away fish, or becomes entangled in nets.** The species are black caiman (C, black bars), giant otter (O, dark grey bars), boto (B, light grey bars) and tucuxi (T, white bars).                

conflict among the study species, and a chi-squared test was used to determine if the response of fishers to entanglement in fishing nets differed among species. We also calculated the nonlinear fluvial distance from the nearest urban centre, Carauari, to each community and the percentage of *várzea* floodplain forest within 5 km of each community, using ArcGIS v 10.2.2 (Esri, Redlands, CA, USA). We then examined the influence of fluvial distance to Carauari and percentage *várzea* forest cover on the eight binary interview questions using binary logistic regression. For the literature review, we used Spearman's rank correlation to investigate the change in the number of reviewed studies over time. A Fisher's Exact Test was implemented to determine if the severity of conflict differs between animal body mass categories (<10 kg, 10–49 kg, ≥50 kg) or the species IUCN Red List category. All data analysis was conducted by PC, using SPSS v 22 (*IBM Corporation, 2014*) and R v 1.4.1106 (*R Core Team, 2021*).

## RESULTS

### Amazonian case study

Interview responses showed a significant difference between the perception of black caiman, giant otters, botos and tucuxis as problem species (Chi-squared: $\chi^2$ = 204.69, df = 3, *p* < 0.001, *n* = 49). Black caiman was consistently regarded as the greatest source of conflicts (mean rank = 1.37), followed by the boto (2.06), giant otter (2.52) and tucuxi (4.00) (*n* = 49). The black caiman was ranked significantly higher as a 'problem species' than the boto (Mann–Whitney: *U* = 573.5, Z = −4.841, *p* < 0.001, *n* = 49), and the boto was ranked significantly higher than the giant otter (*U* = 714.0, Z = −3.195, *p* < 0.001, *n* = 49). Of the 49 interviews conducted, 100% of interviewees reported black caiman and boto as 'problem species', followed by 89.8% for the giant otter (Fig. 3). In the study area, at least nine cases of lethal attacks on humans by black caiman involving both adults and

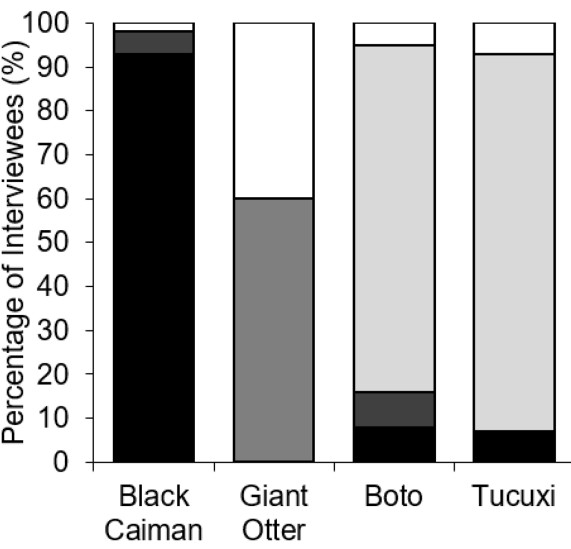

**Figure 4 Percent of interviewees (*n* = 49) who indicated the potential outcomes of entanglement in fishing nets by four species of piscivorous mammals and reptiles.** Outcomes are being killed by fishers (black), dying without fisher intervention (dark grey), being released by fishers (light grey) or escaping without fisher intervention (white).

children have been reported between 2007 and 2020, a rate of about 0.3 persons killed each decade per 1,000 people (C. Peres, 2021, personal communication).

Fishers' responses showed a significant difference among all species regarding the outcomes whenever found entangled in fishing nets ($\chi^2$ = 152.12, df = 9, $p$ < 0.001, $n$ = 49). Black caimans were reported to be killed by 93.0% of fishers (Fig. 4). In contrast, 79.0% of botos and 85.7% of tucuxi were released alive, and 40.0% of interviewees stated that giant otters could escape from gillnets without assistance (Fig. 4). In response to conflicts with the black caiman, 16.7% of interviewees reported that they changed fishing locations as a result, compared to 33.3% for the giant otter, 12.5% for the boto and no interviewees for the tucuxi. Giant otters were reported by 50.0% of interviewees as being responsible for a perceived decline in matrinxã (Brycon cephalus), and 97.9% of interviewees stated that giant otters spatially displace fish (Fig. 3). Most interviewees reported that they could continue coexisting with these four species if populations were to increase in the future, ranging from 60.4% of interviewees who considered that coexistence with the black caiman is possible, to 75.0% for the tucuxi (Fig. S1). With the exception of one variable, no interviewee responses showed a significant relationship with fluvial distance from the nearest urban centre of Carauari or the percentage of *várzea* floodplain forest found within a 5-km buffer area around each community (Table S1). The exception is the damage caused to gillnets by giant otters, which increased with fluvial distance from Carauari (binary logistic regression: β = 0.007, $p$ = 0.009, $n$ = 49).

## Literature review

A total of 143 studies were published between 1962 and 2020, and the number of sources published per year increased over time, including years with no studies reported (Spearman's: $r_s$ = 0.882, $p$ < 0.001, $n$ = 59; Fig. 5). These studies covered 33 countries in the

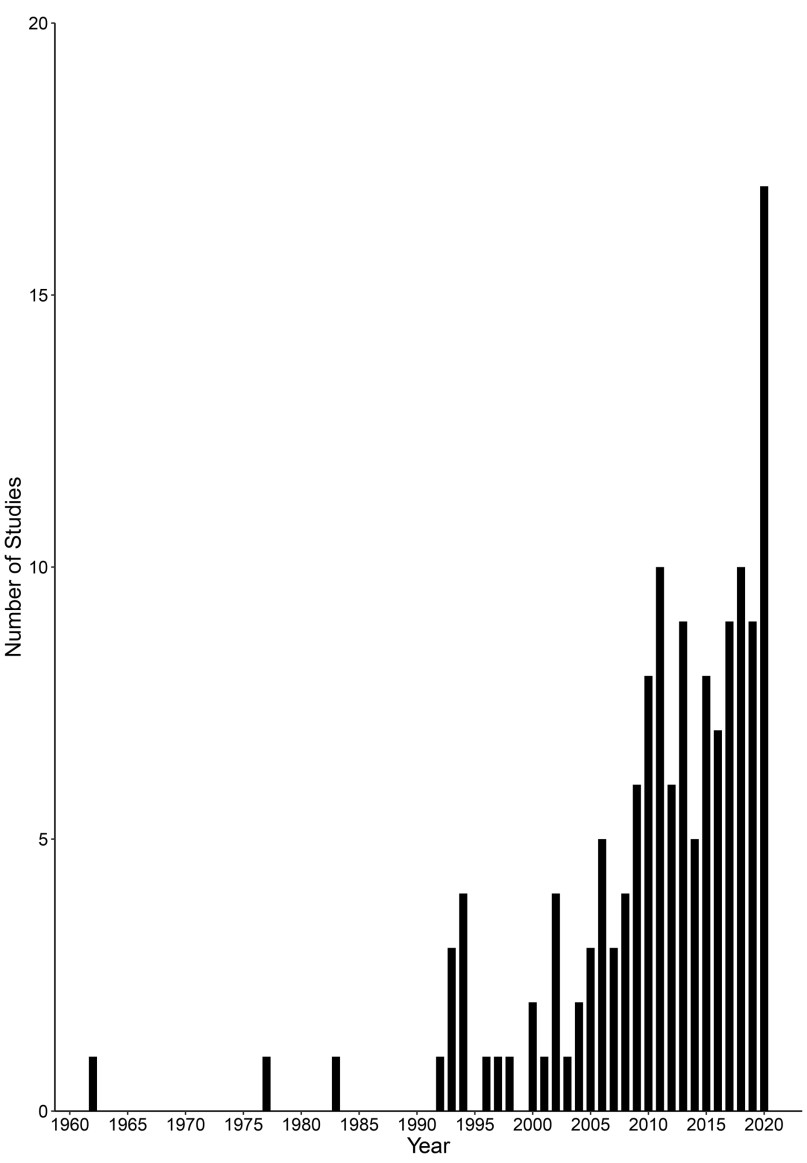

**Figure 5 Number of human-wildlife conflict studies concerning crocodilians, cetaceans and otters over time (1962–2020).**

tropics and subtropics across Africa, Asia, Australia, North America and South America. Brazil was the country with the most studies undertaken ($n = 24$), followed by India ($n = 20$) and Australia ($n = 19$). All other countries had seven or fewer studies.

The compiled studies reported conflicts with 34 species of crocodilians, cetaceans and otters (Table 2). Types of conflicts reported with these species included attacks on humans, depredation of fish, damage to fish nets, competition for economically important fish species and entanglement in fishing equipment.

Attacks on humans accounted for 30.8% of studies (Table 2). The literature included 44 studies concerning sub-lethal or lethal attacks by crocodilians, one study for otters and none for cetaceans. Economic and livelihood impacts such as net damage or competition for fish accounted for 28.7% of studies (Table 2). Otters accounted for 50.0% of the

**Table 2 List of crocodilian, cetacean and otter species from tropical and subtropical regions.** For each species, the frequency of conflict studies (literature coverage category, number of primary literature studies mentioning conflict), frequency of conflict study per type of conflict, severity of the human-wildlife conflict, body mass and IUCN Red List category are listed.

| Species | Frequency of studies[1] | Attacks[2] | Net damage and depredation[2] | Entanglement[2] | Management[2] | Perceptions[2] | Severity of conflict[1] | Body mass category (kg) | IUCN RED List status[6] |
|---|---|---|---|---|---|---|---|---|---|
| Crocodilians | | | | | | | | | |
| African Dwarf Crocodile (*Osteolaemus tetraspis*) | Rare (1) | 0 | 0 | 0 | 0 | 1 | Low | 10–49[3] | VU |
| American Alligator (*Alligator mississippiensis*) | Frequent (6) | 5 | 0 | 0 | 1 | 0 | High | ≥50[3] | LC |
| American Crocodile (*Crocodylus acutus*) | Frequent (5) | 2 | 0 | 1 | 1 | 1 | Severe | ≥50[3] | VU |
| Australian Freshwater Crocodile (*Crocodylus johnstoni*) | Infrequent (2) | 2 | 0 | 0 | 0 | 0 | Moderate | 10–49[3] | LC |
| Black Caiman (*Melanosuchus niger*) | Frequent (5) | 3 | 1 | 1 | 0 | 1 | High | ≥50[3] | LC |
| Broud-snouted Caiman (*Caiman latirostris*) | Rare (1) | 0 | 0 | 0 | 0 | 1 | Moderate | 10–49[3] | LC |
| Central African Slender-snouted Crocodile (*Mecistops leptorhynchus*) | DD (0) | 0 | 0 | 0 | 0 | 0 | DD | Not available | Not assessed |
| Chinese Alligator (*Alligator sinensis*) | DD (0) | 0 | 0 | 0 | 0 | 0 | DD | 10–49[3] | CR |
| Cuban Crocodile (*Crocodylus rhombifer*) | DD (0) | 0 | 0 | 0 | 0 | 0 | DD | ≥50[3] | CR |
| Dwarf Caiman (*Paleosuchus palpebrosus*) | DD (0) | 0 | 0 | 0 | 0 | 0 | DD | ≤10[3] | LC |
| False Gharial (*Tomistoma schlegelii*) | DD (0) | 0 | 0 | 0 | 0 | 0 | High | ≥50[3] | VU |
| Gharial (*Gavialis gangeticus*) | Rare (1) | 0 | 0 | 1 | 0 | 0 | Low | ≥50[3] | CR |

| Species | Frequency of studies[1] | Attacks[2] | Net damage and depredation[2] | Entanglement[2] | Management[2] | Perceptions[2] | Severity of conflict[1] | Body mass category (kg) | IUCN RED List status[6] |
|---|---|---|---|---|---|---|---|---|---|
| Hall's New Guinea crocodile (Crocodylus halli) | DD (0) | 0 | 0 | 0 | 0 | 0 | DD | Not available | Not assessed |
| Morelet's Crocodile (Crocodylus moreletii) | Infrequent (2) | 0 | 0 | 1 | 0 | 1 | High | 10–49[3] | LC |
| Mugger (Crocodylus palustris) | Frequent (5) | 4 | 2 | 0 | 0 | 0 | Severe | ≥50[3] | VU |
| New Guinea Crocodile (Crocodylus novaeguineae) | DD (0) | 0 | 0 | 0 | 0 | 0 | DD | 10–49[3] | LC |
| Nile Crocodile (Crocodylus niloticus) | Frequent (14) | 9 | 6 | 0 | 0 | 1 | Severe | ≥50[3] | LC |
| Orinoco Crocodile (Crocodylus intermedius) | DD (0) | 0 | 0 | 0 | 0 | 0 | Moderate | ≥50[3] | CR |
| Philippine Crocodile (Crocodylus mindorensis) | DD (0) | 0 | 0 | 0 | 0 | 0 | Moderate | 10–49[3] | CR |
| Saltwater Crocodile (Crocodylus porosus) | Frequent (33) | 17 | 1 | 0 | 13 | 2 | Severe | ≥50[3] | LC |
| Siamese Crocodile (Crocodylus siamensis) | DD (0) | 0 | 0 | 0 | 0 | 0 | Moderate | 10–49[3] | CR |
| Slender-snouted Crocodile (Mecistops cataphractus) | Rare (1) | 0 | 0 | 0 | 0 | 1 | Low | ≥50[3] | CR |
| Smooth-fronted Caiman (Paleosuchus trigonatus) | Infrequent (2) | 0 | 1 | 0 | 0 | 2 | Low | ≤10[3] | LC |
| Spectacled Caiman (Caiman crocodilus) | Frequent (6) | 3 | 1 | 1 | 0 | 2 | High | 10–49[3] | LC |

(Continued)

| Species | Frequency of studies[1] | Attacks[2] | Net damage and depredation[2] | Entanglement[2] | Management[2] | Perceptions[2] | Severity of conflict[1] | Body mass category (kg) | IUCN RED List status[6] |
|---|---|---|---|---|---|---|---|---|---|
| West African Crocodile (*Crocodylus suchus*) | Infrequent (2) | 0 | 0 | 0 | 0 | 2 | Low | Not available | Not assessed |
| Yacaré (*Caiman yacare*) | Rare (1) | 0 | 0 | 0 | 0 | 1 | Moderate | 10–49[3] | LC |
| Otters | | | | | | | | | |
| African Clawless Otter (*Aonyx capensis*) | Frequent (5) | 0 | 4 | 0 | 0 | 1 | Low | 10–49[4] | NT |
| Asian Small-clawed Otter (*Aonyx cinereus*) | Infrequent (2) | 0 | 2 | 0 | 0 | 0 | Low | ≤10[4] | VU |
| Congo Clawless Otter (*Aonyx congicus*) | Infrequent (2) | 0 | 2 | 0 | 0 | 0 | Low | 10–49[4] | NT |
| Eurasian Otter (*Lutra lutra*) | DD (0) | 0 | 0 | 0 | 0 | 0 | DD | 10–49[4] | NT |
| Giant Otter (*Pteronura brasiliensis*) | Frequent (11) | 0 | 9 | 0 | 0 | 2 | Low | 10–49[4] | EN |
| Hairy-nosed Otter (*Lutra sumatrana*) | DD (0) | 0 | 0 | 0 | 0 | 0 | DD | ≤10[4] | EN |
| Neotropical River Otter (*Lontra longicaudis*) | Frequent (7) | 0 | 6 | 0 | 0 | 1 | Low | ≤10[4] | NT |
| North American River Otter (*Lontra canadensis*) | Rare (1) | 1 | 0 | 0 | 0 | 0 | Moderate | 10–49[4] | LC |
| Smooth-coated Otter (*Lutrogale perspicillata*) | Frequent (6) | 1 | 5 | 0 | 0 | 1 | Low | ≤10[4] | VU |
| Spotted-necked Otter (*Hydrictis maculicollis*) | Frequent (6) | 0 | 5 | 0 | 0 | 1 | Low | ≤10[4] | NT |
| Cetaceans | | | | | | | | | |
| Amazon River Dolphin (*Inia geoffrensis*) | Frequent (14) | 0 | 4 | 9 | 0 | 1 | Low | ≥50[5] | EN |
| Baiji (*Lipotes vexillifer*) | Infrequent (4) | 0 | 0 | 4 | 0 | 0 | Low | ≥50[5] | CR |
| Table 2 (continued) | | | | | | | | | |
|---|---|---|---|---|---|---|---|---|---|
| Species | Frequency of studies[1] | Attacks[2] | Net damage and depredation[2] | Entanglement[2] | Management[2] | Perceptions[2] | Severity of conflict[1] | Body mass category (kg) | IUCN RED List status[6] |
| Irrawaddy Dolphin (*Orcaella brevirostris*) | Infrequent (4) | 0 | 0 | 3 | 0 | 0 | Low | ≥50[5] | EN |
| South Asian River Dophin (*Platanista gangetica*) | Frequent (14) | 0 | 1 | 12 | 0 | 1 | Low | ≥50[5] | EN |
| Tucuxi (*Sotalia fluviatilis*) | Frequent (6) | 0 | 4 | 2 | 0 | 0 | Low | 10–49[5] | EN |
| Yangtze Finless Porpoise (*Neophocaena asiaeorientalis* ssp. *asiaeorientalis*) | Infrequent (4) | 0 | 0 | 4 | 0 | 0 | Low | ≥50[5] | CR |

Notes:
[1] See Table 1 for description of categories; DD, Data Deficient.
[2] A study was not mutually exclusive to any single type of conflict.
[3] *Lakin et al. (2020)*.
[4] *Hunter (2011)*.
[5] *Macdonald (2009)*.
[6] Conservation status according to the Red List categories of the International Union for Conservation of Nature (IUCN): LC, least concern; NT, Near Threatened; VU, Vulnerable; EN, Endangered; CR, Critically Endangered.

species documented in the literature under this category, with the giant otter the most cited species accounting for 18.4% of these studies. The Nile crocodile (*Crocodylus niloticus*) (8.2% of studies) and boto (8.2%) were the most cited crocodilian and cetacean respectively. Studies focused on entanglement in fishing nets comprised 19.6% of the literature (Table 2). These studies covered five species of crocodilian and six species of cetaceans, the latter accounting for 87.1% of all studies on entanglement. Management or resolution of conflict was the subject of only 10.5% of all studies covering five countries, notably Australia which accounted for 60.0% of management studies (Table 2). Perceptions of conflict accounted for 9.8% of studies, across 15 crocodilians, eight otters and seven cetaceans (Table 2).

The frequency of reported conflicts varied across species; 35.7% of species were frequently documented, 21.4% infrequently documented, 14.3% rarely documented and 28.6% listed as data deficient with no evidence or documentation of conflict occurring (Table 2). Over half (50.3%) of all studies included reports of conflicts with crocodilians, while approximately a quarter of studies reported conflicts for cetaceans (26.6%) or otters (23.1%) (Table 2). Five species accounted for nearly half (49.7%) of all studies: the saltwater crocodile (*Crocodylus porosus*) (33 studies), Amazon river dolphin (14), Nile crocodile (14), South Asian river dolphin (*Platanista gangetica*) (14) and the giant otter (11). Six species listed as either endangered or critically endangered on the IUCN Red List had no documentation of conflict occurring.

**Table 3 Number of species per IUCN Red List status ($n = 33$) and body mass category ($n = 34$) in each conflict severity category.**

| | Conflict severity category[1] | | | |
|---|---|---|---|---|
| | Low | Moderate | High | Severe |
| IUCN Red List Status[2] | | | | |
| CR | 4 | 3 | 0 | 0 |
| EN | 5 | 0 | 0 | 0 |
| VU | 3 | 0 | 1 | 2 |
| NT | 4 | 0 | 0 | 0 |
| LC | 1 | 4 | 4 | 2 |
| Body Mass Category (kg) | | | | |
| ≤10 | 5 | 0 | 0 | 0 |
| 11–49 | 5 | 6 | 2 | 0 |
| ≥50 | 7 | 1 | 3 | 4 |

**Notes:**
[1] See Table 1 for description of categories.
[2] Conservation status according to the Red List categories of the International Union for Conservation of Nature (IUCN): LC, Least Concern; NT, Near Threatened; VU, Vulnerable; EN, Endangered; CR, Critically Endangered.

Only crocodilians were assigned to the severe and high conflict categories, with four and five species in each respectively. The North American river otter (*Lontra canadensis*) was the only non-crocodilian species to be assigned a conflict severity category of moderate. All other otters and cetaceans were classified as low conflict severity. The severity of conflict differed significantly among body mass categories, with larger species more likely to be involved in severe conflicts (Fisher's Exact Test: $p < 0.05$; Table 3). The severity of conflict also differed significantly among IUCN Red List categories, with species classified in less threatened categories displaying higher levels of conflict (Fisher's Exact Test: $p < 0.01$; Table 3).

# DISCUSSION

## Amazonian case study

We found that interviewee responses displayed significant differences between the perceptions of black caiman, giant otter, boto and tucuxi as 'problem species'.
The consistent identification of black caiman as the highest-ranking problem species reflects the level of direct threat to human life that the black caiman poses, and most adults in the middle-third of the Juruá River basin know, or have heard of someone, who has been killed by a black caiman within their lifetimes. This finding was in strong agreement with our literature review where all the case study species except the black caiman were classified as low severity. That the differences in conflict severity between the giant otter, boto and tucuxi reported in our case study were not also detected in the literature review reflects our focus on attacks in the latter. Despite these existing conflicts, most interviewees reported that they could continue living in proximity with these four species, even if their populations were to increase in the future. The percentage of interviewees

agreeing with this statement reflected the problem ranking status, being lowest for the black caiman.

Fishers' typical responses to entanglement in fishing nets of the four study species differed, with black caiman typically killed, botos and tucuxis released, and the giant otter often dying before being discovered. This can partly be explained by the differing ability of each species to escape entanglement in nets, with 40.0% of interviewees stating that giant otters could escape (Fig. 4). This value is much higher than for the black caiman, boto or tucuxi, reflecting the ability of giant otters to tear through nets with their teeth and dexterous paws. For the boto, our findings differ from reports in other areas, such as the Central Amazon where this species is intentionally killed as bait for piracatinga catfish (*Calophysus macropterus*) fisheries (*Loch, Marmontel & Simões-Lopes, 2009*; *Alves, Zappes & Andriolo, 2012*). The positive relationship that we found between damage to gill-nets by the giant otter and distance from the nearest town, is challenging to explain. It is likely that giant otter abundance increases at greater distances from urban centres, particularly given historic range wide declines due to hunting pressure (*Antunes et al., 2016*), which could result in more frequent depredation of fish further from towns. However, without reliable survey data for the giant otter population in the study area it is not currently possible to confirm this.

We found no relationship between interviewee responses and percentage of *várzea* floodplain forest found within a 5-km buffer area around each community. This was in contrast to our expectation that greater access to aquatic habitat would raise conflict severity by increasing the proximity and potential for interactions between humans and the study species. One possible explanation for the absence of a clear relationship here could be the additional role of seasonal variation. For example, conflict occurrence may be higher in the wet season, when water levels rise allowing aquatic species access to the flooded forest (*Junk et al., 2011*). There is some evidence that conflict with giant otters is highest during the wet season, when fish disperse into the flooded forest and otters become more generalist, targeting vulnerable species (*Cabral et al., 2010*; *Rosas-Ribeiro, Rosas & Zuanon, 2012*). This happens to coincide with the time of year when the income for fishers tends to be lowest (*Junk, 1984*; *Cabral et al., 2010*; *Rosas-Ribeiro, Rosas & Zuanon, 2012*), which could exacerbate any potential conflicts.

## Fish depredation and net damage

The growing spatial overlap between humans, crocodilians, cetaceans and otters has in some locations increased negative interactions. This may result in attacks on people, economic losses or coincidental declines in fish stocks, with these species potentially blamed or persecuted even if overfishing is the driving factor (*Gopi & Pandav, 2009*; *Recharte, Bowler & Bodmer, 2009*; *Fukuda, Manolis & Appel, 2014*; *Lima, Marmontel & Bernard, 2014a*).

Economic losses from fish depredation and net damage are often cited in the literature but studies that provide quantifiable, standardised data and solutions to resolve them are limited (*Aust et al., 2009*). Levels of depredation are currently far better compiled for marine ecosystems, together with proposed methods to reduce depredation (*Tixier*

*et al., 2021*). Methods for reducing predation in terrestrial ecosystems, *e.g.* of felids on pastoral livestock, are also better documented than for freshwater fisheries, although rigorous evaluation of these proposals remains poor (*Inskip & Zimmermann, 2009*). However, depredation of fish stocks and net damage by aquatic animals can still be extensive (*Aust et al., 2009*; *Barbieri et al., 2012*). Our literature review confirmed that crocodilians, cetaceans and otters from across the tropics and subtropics are all reported to depredate fish, damaging nets in the process and cause competition with commercial and subsistence fisheries by catching commercially valuable fish and displacing fish.

Riverine communities in tropical and subtropical regions worldwide often depend on fish for both dietary protein and financial income, and damage to nets can therefore severely impact their livelihoods (*Michalski et al., 2012*). Our case study found the black caiman, giant otter and boto all damage fishing equipment in Amazonia and, although we did not document the frequency or economic severity of this damage in the Juruá, black caiman have been reported to damage up to 50.0% of commercially deployed gill nets elsewhere in the Amazon (*Peres & Carkeek, 1993*). Replacement of fishing nets is likely to have major financial ramifications for fishers with limited resources, as in Namibia where approximately 71,500 nets are damaged annually by the Nile crocodile and the purchase of new nets can often exceed monthly income (*Aust et al., 2009*).

Compared to crocodilians, depredation by otters, mostly involves the giant otter, particularly in relation to matrinxã fisheries in the western Amazon, which are important for both subsistence and trade (*Santos, Ferreira & Zuanon, 2006*; *Rosas-Ribeiro, Rosas & Zuanon, 2012*). Perceived competition can lead to retaliatory killing of 'problem individuals' which has detrimental impacts on the species by reducing population recovery (*Brum et al., 2021*), and resource depletion of fisheries can further intensify competition and conflict. Economic losses due to cetaceans are documented in both South America and Asia (*Kelkar et al., 2010*; *Alves, Zappes & Andriolo, 2012*; *Campbell et al., 2020*). There are consistent reports of the boto raiding and damaging nets, with all interviewees in our case study supporting findings from elsewhere in the Amazon (*Alves, Zappes & Andriolo, 2012*; *Campbell et al., 2020*). This contrasts sharply with the sympatric tucuxi, reflecting the greater levels of animosity towards the boto and potential differences in foraging strategy between these two species (*Martin & Da Silva, 2004*; *Alves, Zappes & Andriolo, 2012*).

## Entanglement

Entanglement in fishing equipment is reported most often in the literature for cetaceans but also threatens crocodilians and otters, with impacts ranging from injury to death across all groups (*Platt & Thorbjarnarson, 2000*; *Choudhary et al., 2006*; *Alves, Zappes & Andriolo, 2012*). All four species in our Amazonian case study were reported by local fishers to become entangled in nets. Specific net types, such as seine nets or nets with polyamide threads, may increase the risk of fatality from entanglement for both otters and cetaceans (*Leatherwood & Reeves, 1994*; *da Silva & Best, 1996*; *Lima, Marmontel & Bernard, 2014b*).

Bycatch through entanglement may be as important for population trends in aquatic animals as for those in the marine realm (*Mangel et al., 2013*; *Anderson et al., 2020*; *Tixier et al., 2021*). For example, entanglement is cited as the primary source of conflict and a contributor to population declines for both the boto and tucuxi (*Campbell et al., 2020*), with the latter now listed as endangered on the IUCN Red List (*da Silva et al., 2018*; *IUCN, 2020*; *Brum et al., 2021*). This has resulted in all river dolphin species worldwide now being listed as endangered, critically endangered or extinct (*IUCN, 2020*). In extreme circumstances entanglement can even contribute to extinction (*Jaramillo-Legorreta et al., 2019*), such as the case of the Yangtze river dolphin (*Lipotes vexillifer*) where 40% of fatalities during the 1990s were attributed to this factor (*Zhou et al., 1998*; *Zhang et al., 2003*; *Turvey et al., 2007*). Techniques to prevent entanglement, such as acoustic deterrent pingers, have been developed for use in marine ecosystems and their effectiveness for freshwater cetaceans has now begun to be tested (*Waples et al., 2013*; *Prajith, Das & Edwin, 2014*; *Snape et al., 2018*; *Tixier et al., 2021*; *Zanon, 2021*).

Human responses to entanglement vary from immediate release to retaliatory killing, and are often influenced by local perceptions and economics (*Sinha, 2002*; *Alves, Zappes & Andriolo, 2012*; *Campbell et al., 2020*). In our case study, we found that the response between the four studies species varied greatly, with black caiman often killed and exploited as a food source. The probability of being killed rather than released likely reflects the level of local conflict severity for that species, which can be influenced by economic and political situations (*Loch, Marmontel & Simões-Lopes, 2009*; *Alves, Zappes & Andriolo, 2012*). In our Juruá waterscape where piracatinga fisheries are not of commercial importance, both the boto and tucuxi were always reportedly released. The situation in Peru is more complex, with most fishers releasing entangled botos and tucuxis, but some ports displaying a higher frequency of use for bait (*Campbell et al., 2020*).

## Body mass and conflict severity

The positive relationship that we found between body mass and conflict severity supports the similar finding of *Inskip & Zimmermann (2009)*, who examined conflict severity with felids, emphasising the need for greater conservation attention on larger bodied species. Our results also agree with previous studies that find large-bodied species or individuals, such as male crocodilians, are often engaged in more severe conflicts and represent a greater threat to human life (*Caldicott et al., 2005*; *Campbell et al., 2013*; *Fukuda et al., 2015*). The strong pattern in our results can also be considered a conservative assessment, as we used average female body mass for crocodilians, from *Lakin et al. (2020)*, rather than maximum reported body mass. Using maximum body mass would exaggerate the strength of the relationship, as male crocodilians can achieve much larger body masses.

A possible limitation of our method is the focus on attacks on humans, as this trend is largely influenced by crocodilians. For instance, of the 26 recognised crocodilian species, 15 have been documented to attack humans, seven of which were responsible for lethal attacks (*Sideleau & Britton, 2013*). In comparison, otters rarely attack humans, with 95.2% of documented cases in *Belanger et al. (2011)* linked to the North American river

otter, and we could find no cases of cetacean attacks. We were unable to quantify other types of conflict such as entanglement rate or fish depredation as part of a repeatable method to calculate conflict severity. For some species, such as the boto, this may have led to assigning a low conflict severity category overall despite the occurrence of more severe conflicts in some localities (*Loch, Marmontel & Simões-Lopes, 2009*; *Alves, Zappes & Andriolo, 2012*).

## IUCN Red List status and conflict severity

We found species in lower threat categories on the IUCN Red List displayed higher levels of conflict with humans. This finding has important management considerations for species found in lower threat categories, or those that are recovering in population size and increasingly experiencing conflict. Coexistence with recovering species can depend on perceived or actual population trends, and the conservation strategies used to manage conflict (*Brackhane et al., 2018*; *Fukuda et al., 2019*; *Patro & Padhi, 2019*; *Fukuda et al., 2020*). For instance, in our Amazonian case study, where black caiman and giant otter populations have increased following declines from peak levels of historical hunting (*Antunes et al., 2016*), 14.6% of interviewees reported that coexistence with these species depends on their future population trends (*Lima, Marmontel & Bernard, 2014a*; *Pimenta et al., 2018*; *Marioni et al., 2021*).

The relationship we found between threatened status and conflict severity is likely driven by the studies on crocodilians, with a high number of studies on common and widespread species, but with several rarer species poorly studied and/or with no documentation of conflicts. Threatened species across all taxonomic groups may suffer a bias in reduced reporting of conflicts as they occupy limited ranges or occur in particular countries (*Sideleau & Britton, 2013*). The approach adopted in our study is suitable for investigating broad scale patterns across species at a large-scale but is less likely to identify differences in conflict severity that may influence species populations at the local level, such as the higher severity of saltwater crocodile conflicts in Timor-Leste compared to Australia (*Fukuda, Manolis & Appel, 2014*; *Brackhane et al., 2018*; *Fukuda et al., 2020*). We therefore advise caution in the interpretation of some aspects of our results for such threatened species, as even minimal conflict could have a disproportionately higher effect due to their inherent rarity.

## Research needs and limitations

In our case study, we used binary question answers to assess conflicts, but a more quantitative approach *e.g.* quantifying the financial cost of net damage to each fisher could have improved the scope for analysing different predictors of conflict severity. Anecdotally, from our review we noticed that few studies quantify conflicts, as with our case study, but such quantification would improve the value and impact of studies. Increased availability of data in the literature regarding the occurrence of entanglement in fishing nets, and economic losses due to net damage, would have strengthened the analyses

possible from our review. In the absence of such data, we necessarily defined conflict severity based solely using attacks. Such an approach did not quantify other forms of conflict which could still generate high conflict severity and have potential impacts for both the conflict species and fishers. This will have biased our measure of severity towards crocodilians and therefore had an impact on our broad scale analysis.

Our review indicates that conflicts with the majority of crocodilians, cetaceans and otters have been infrequently documented in the literature and there is a need for wider systematic reporting, particularly for species experiencing high levels of conflict. Our focus on the frequency of studies, which is an approximate measure of how well studied a species or topic might be and does not take into account the quality or impact of each study. Our exclusion of grey literature and non-English language publications, and our decision not to conduct searches using Web of Science, may also have precluded additional studies being located (*Haddaway et al., 2015*) and led to the possible omission of some useful information (*Inskip & Zimmermann, 2009*). Despite these potential limitations, the quantitative assessment in our study provides a valuable contribution to address the poor understanding of human-wildlife conflicts in aquatic systems both in the Amazon and across the tropics.

## CONCLUSIONS

Simultaneously conducting our case study and literature review has allowed us to identify local and broad scale patterns of human-wildlife conflict with crocodilians, cetaceans and otters. Multiple factors may influence conflict severity including species body mass or rarity, but better estimations of conflict severity are required, by integrating forms of conflict other than attacks into calculations. The dual nature of our research allows us to make the following recommendations for future research in aquatic systems to better prioritise conflict resolution efforts. Our main recommendations for field studies are to: (1) quantify the economic costs of conflicts, for example damage to gill-nets, at the fisher level; (2) identify the types of fishing equipment used; (3) quantify the entanglement rate per fisher, for different types of fishing equipment; and (4) directly quantify the attack rate per fisher. Future research at a broader scale should focus on (1) determining the severity of any conflicts for species identified here as infrequently or rarely documented, with priority given to those identified as experiencing a more severe level of conflict; (2) assessing conflict resolution techniques to determine their effectiveness; (3) conducting a meta-analysis of economic losses due to net damage and fish depredation; and (4) conducting a meta-analysis of entanglement in fishing equipment.

## ACKNOWLEDGEMENTS

We are grateful to Franciney Silva da Souza for assisting fieldwork and to all reserve residents for their hospitality and participation in interviews. We wish to thank Frank Hajek and Jessica Groenendijk for allowing us to use their photographs of black caiman and giant otter, and Sannie Brum for photographs of the boto and tucuxi. This publication is part of the Instituto Juruá series (www.institutojurua.org.br).

### Funding
This study was funded by a DEFRA Darwin Initiative grant (Ref. 16-001) awarded to Carlos A Peres, a CAPES PhD scholarship (Ref. 1144985) and CAPES postdoctoral grant (Ref. 1666302) to João Vitor Campos-Silva, and a CAPES postdoctoral grant (Ref. 1530532) and internal funding from Anglia Ruskin University to Joseph E. Hawes. The funders had no role in study design, data collection and analysis, decision to publish, or preparation of the manuscript.

### Grant Disclosures
The following grant information was disclosed by the authors:
DEFRA Darwin Initiative Grant: 16-001.
CAPES PhD Scholarship: 1144985.
CAPES Postdoctoral Grant: 1666302 and 1530532.
Anglia Ruskin University.

### Competing Interests
The authors declare that they have no competing interests.

### Author Contributions
- Patrick Cook conceived and designed the experiments, performed the experiments, analyzed the data, prepared figures and/or tables, authored or reviewed drafts of the paper, and approved the final draft.
- Joseph E. Hawes conceived and designed the experiments, authored or reviewed drafts of the paper, and approved the final draft.
- João Vitor Campos-Silva conceived and designed the experiments, authored or reviewed drafts of the paper, and approved the final draft.
- Carlos A. Peres conceived and designed the experiments, authored or reviewed drafts of the paper, and approved the final draft.

### Human Ethics
The following information was supplied relating to ethical approvals (*i.e.*, approving body and any reference numbers):

Fieldwork, including all ecological and cultural data collection activities, was authorised by the Ministério do Meio Ambiente/Instituto Chico Mendes de Conservação da Biodiversidade of the Brazilian government (permit number: 45054-1). None of our interview questions, which were about human-wildlife conflicts, solicited sensitive or controversial information from our local informants, who voluntarily provided information and gave verbal consent.

## Field Study Permissions

The following information was supplied relating to field study approvals (*i.e.*, approving body and any reference numbers):

Fieldwork was authorised by the Ministério do Meio Ambiente/Instituto Chico Mendes de Conservação da Biodiversidade of the Brazilian government (permit number: 45054-1).

## Data Availability

The raw data is available in the Supplemental File.

## Supplemental Information

Supplemental information for this article can be found online at http://dx.doi.org/10.7717/peerj.12688#supplemental-information.

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
