# Peer review of "Human-wildlife conflicts with crocodilians, cetaceans and otters in the tropics and subtropics"

_PeerJ, doi:10.7717/peerj.12688_

## Round 0.1 · original submission · Major Revisions

Overview
This manuscript includes two distinct investigations. One is a literature review of human-wildlife conflicts with freshwater mammals and reptiles. The other is an interview-based study of conflicts with four species (otters, two cetaceans, caiman) in a specific region of the upper Amazon. The literature review identified 141 references to 30 species in 33 countries between 1977 and 2020. Less than one third were considered well documented. There was an association (direction not stated) between IUCN Red List category and severity of conflict but not between size (body mass) and severity of conflict. An additional analysis of lethal and sub-lethal attacks on humans by crocodilians highlighted the importance of saltwater and Nile crocodiles in such attacks, although attacks by a total of 17 species had been documented. The Amazonian study indicated that caiman were considered the most problematic species, followed by the boto, giant otter, and tucuxi. Most caiman trapped in nets were killed whereas the two cetaceans (boto, tucuxi) were mostly released, and otters typically freed themselves from nets. Most interviewees believed that coexistence with all species was possible. Only otter damaged showed a relationship with remoteness from town. There were no comparisons of the results to other systems such as forests or inshore marine or temperate freshwater, and conclusions seemed mostly related to the need for more studies and better data.

Both reviews were generally positive suggesting the need for only minor revisions. However, Reviewer 1 identified possible gaps in the literature review. Reviewer 2 indicated that a clearer justification of the research questions and an additional focus on negative implications of conflict for wildlife would be helpful, in addition to a need to better integrate that two studies. My concerns overlap with those of the reviewers but are somewhat greater. Although both parts of the manuscript address the topic of conflict with tropical/sub-tropical freshwater piscivores, I do not find a strong logical link between them; this must be strengthened to justify inclusion in the same manuscript. The questions asked by each study and the justification of these questions need elaboration. I am not confident that the methods of the literature review are sufficiently well described to be repeated by another researcher. The Discussion needs greater focus; at present, it touches on most of the findings rather briefly or not at all and raises topics not directly related to the findings. It is difficult to identify specific conclusions arising from this research.

The writing is generally clear and error free. I have provided an annotated pdf with some minor corrections indicated. All my substantial comments are stated below. You should treat these comments as if they were a third review, i.e., report the changes you made if the comments are valid and explain why if they are not valid. I apologize for the rather long-winded nature of these comments.

Note that Reviewer 2 has also provided an annotated pdf. This includes both minor comments on word use and grammar as well as much more substantial comments. You should copy the substantial comments into your rebuttal letter and respond to them specifically. The minor comments on grammar and word use by the reviewer and me do not have to be responded to specifically, unless you disagree and are retaining the original text.

Editor’s Comments

Major concerns

1. Integration of the two studies
What is the logical link between a world-wide literature review and the results of interviews regarding conflict with four piscivorous species in a specific region of the Amazon? For example, was the review undertaken to provide context for the Amazon study or was the Amazon study undertaken to address some of the limitations of the literature raised by the review? If the studies do not have a logical connection, it would be better to publish them as separate papers for logical coherence and to make the contributions of each study clear.

2. Objectives of the literature review
• The present objectives (L66-71) do not make it clear which are to be met by the review and which by the interviews. Rather than integrating the studies, this adds confusion because some questions could only be addressed by one of the studies (e.g., Red List correlation, correlation with amount of varzea).
• The objective about ‘key factors inducing conflicts’ is very vague. What kind of key factors are you investigating?
• All objectives should flow logically from the preceding Introduction which should include a review of relevant background. Terms used in the objectives should have been explicitly introduced previously. The lack of a clear introduction to the questions was noted by a Reviewer with regard to some of the specific objectives.
• It seems to me that some questions you considered, at least partially, are not included in the list. Objectives will make more sense to readers if they show a logical progression. For example, some objectives that could be more explicitly stated in a logical order include:
1. What types of conflicts occur with freshwater piscivores?
2. How many and which species are involved in each type of conflict?
3. How is severity of conflict related to the type of conflict and the species involved?
4. Does species size or Red List status predict conflict severity?
• If identifying information gaps is a specific objective, you should indicate in the Introduction what information is needed. Are there specific types of information that you are looking for, for example based on literature from other systems? If you conducted a systematic search for certain types of information so that you can be certain it was rare or not present, this would be an appropriate objective. Otherwise, you could simply mention in the Discussion types of information that you noticed to be lacking in your review.

3. Methods of the literature review
• You state L63-64 that the topic of the literature review is the conflict between fisheries and tropical and sub-tropical freshwater piscivores worldwide. This scope needs to be defined more clearly and the limits justified.
o As a reviewer notes, it is not clear why you don’t include this birds, even though conflicts can occur with this group.
o Why exclude fishes and amphibians as freshwater piscivores? Large or spiny fishes may also cause problems with gill nets, for example. Piranhas, sting rays, and electric eels may also cause problems for fishers. Cetopsid catfishes may consume larger fish caught in gill nets. I would not be surprised if freshwater sharks also cause problems in some places.
o Is the focus on crocodilian reptiles due to the distribution of conflicts or a priori taxon selection? For example, are turtles and large or venomous snakes never a problem?
o Why limit the survey to piscivores? It seems possible that non-piscivores could affect fisheries and would certainly be affected by them.
o Why exclude aquaculture? Is this not a freshwater fishery? There is certainly a literature on avian and piscine predators in this area, including mitigation.
o Why include attacks on livestock, bathers and swimmers? These do not involve fisheries.
o If you are going to include attacks on livestock, did you include or exclude pets? (Lots of news reports of alligator attacks on pets in Florida.)
o Are you considering only some fishing methods (gill nets) to the exclusion of others such as spearing, hook-and-line, seining and trapping?
o You mention the negative conservation impacts in marine systems (L60) but do not seem to include bycatch issues in your review.
o Why is your focus only on tropical and sub-tropical regions? I think this could be justified, but you should be explicit about why. Also, should you define the latitudes or how you decided if a case was sub-tropical or temperate?
o Did you also exclude studies in which a species was in the marine environment? What about brackish water? (Lots of mention of saltwater crocodiles, which like some other crocodiles, are not restricted to freshwater.)
o Why, in relation to your goals, did you include the CrocBITE data?
• The literature search method needs to be more clearly defined so that another researcher could repeat the study and so that readers can check for its validity.
o Do you need to be more explicit about the Boolean operators (where you used ‘and’ vs. ‘or’)?
o As a reviewer noted, by including species names, you may have limited your search to species you already knew may cause problems. Did this limit your capacity to find unfamiliar examples of conflicts?
o By implication, it seems as though you did not search the reference section of papers you found for earlier cited references. This is normal practice in literature reviews that are trying to be thorough, especially because topic-related searches are relatively recent, excluding earlier research from database searches. It seems unlikely that there were no relevant studies before 1977 (L147). How thorough was your search?
o Many literature searches and meta-analyses use Web of Science. Is the coverage of Google Scholar and Scopus likely to be as complete?
o What were the sources for your body mass values? (I later saw that these are noted in the table, but they should be included in Methods.)
o What were the criteria for including/excluding studies from CrocBite?
o Do you have permission to publish the CrocBITE data? Their website states, “You may download a limited sample of this data for research purposes. Any other re-use is strictly prohibited without express written permission. See Access Policy for full details.”
o Also, 2013 seems an odd citation year for the CrocBITE website since this is when the website started. I may be wrong about this. Please check author guidelines for citing a website.
• Categorizing the severity and extent of conflict
o The categories for severity and extent of knowledge are an important part of the Methods and should be included in the main manuscript so that readers clearly understand what they mean.
o Several criteria are ambiguous and need to be clarified.
 For ‘Poorly Documented’, it is not clear how the secondary literature relates to the fewer than 5 primary sources.
 For ‘Research Required’, the supposed criteria simply rephrase the category. It needs to be clear what type of information would allow a species to be included but be insufficient to be categorized as ‘Poorly Documented’.
 It seems a bit odd that a species with one primary source that provided detailed, specific information on many cases would be considered as poorly documented while a species with 6 anecdotal observations in the primary literature would be considered well documented. If this categorization is necessary for quantification, it would be relevant to the Discussion to consider the validity of this categorization.
 For severity, the criteria are not sufficiently clear or grammatically expressed. It took me a while to work out that ‘Severe’ means frequent effects on fisheries plus attacks on people, ‘High’ means frequent effects on fisheries but fewer attacks on people, ‘Moderate’ means fewer effects on fisheries and fewer attacks on people, ‘Low’ means fewer effects on fisheries and no attacks on people. It is not clear how a case in the categories ‘None’ and ‘Data Deficient’ would even be included in your survey. You provided numerical categories for attacks on people but no operational definition for ‘high’, ‘some’ and ‘infrequent’ as applied to incidents of fishery damage.
 Remember that another researcher should be able to repeat your study. It would be desirable to have more than one person assign categories and provide a measure of inter-observer consistency.
 It seems to me that combining categories of impact on fisheries and attacks on people creates an incomplete and ambiguous list (no category for high damage to fisheries but no attacks on people or for frequent attacks on people but no damage to fisheries). Consider separating damage to fisheries and injury to people. I suspect this would make the review substantially more useful.

4. Discussion
• Although Methods and Results separate the literature review and Amazon survey, there is no clear distinction in the Discussion. I would have expected the Discussion to address each of the data sets separately before integrating them into a broader picture. If you decide to retain the sub-categories of the Discussion (Attacks on humans, Economic loss), it should be possible to consider each section separately and then how they relate to each other. Perhaps economic loss will be too general if you focus more clearly on the different types of conflict.
• For each of the data sets, the Discussion should start with the findings of your study and the strength of the patterns. For example, consider the potential for biases in the reporting, publication, and search of the literature. Consider the strength of any patterns observed as well as the strength of lack of patterns. Could sample size or categorical designations or power of the statistical tools explain the pattern?
• Then, you can discuss the relationship among answers, e.g., how the frequency of problems or potential for injury relates to probability of killing vs. releasing. That is, try to integrate and find patterns in your whole data set.
• After clarifying what patterns you have found, relate them to the previous literature, including highlighting what is novel about your findings. Would there be value in comparing your literature survey results to similar surveys done for marine, temperate freshwater and terrestrial predators?
• Minimize discussion of topics unrelated to your specific studies. For example, your studies did not investigate factors influencing attacks, so the extended discussion (L204-212) of this, in contrast to the objectives of your survey is inappropriate. The sub-section on Community, Culture and Conflicts does not appear to be related to any of your findings, and the sub-section on Population Recovery is only marginally related.
• The Conclusions section is a bit long but appropriately highlights future directions as designated in the PeerJ Instructions to Authors. However, topics such as systematic reporting could be raised in the discussion of data reliability and then returned to here. Do you consider your Amazon survey to be an example of the desired level of reporting, or did you learn things in conducting and analyzing the survey that you could recommend to future investigations? The noting of relatively few well documented cases could be a bit misleading when all this means by your criteria is that there are fewer than 6 published examples rather than the quality of documentation.

Other concerns

Abstract
The present abstract provides no results from the Amazon survey and includes interpretations unrelated to the actual data obtained in the literature survey. The final conclusion sentence more or less repeats assertions from the Conclusions but is not based on a considered logical argument for their validity. The Abstract should be carefully restructured after revisions to Results and Discussion.

Introduction
L66. Did your survey actually investigate how the conflicts arise or simply the nature of the conflict?

Results
L158. Here and elsewhere, give the test statistic and d.f. or n each time you report a p-value.
L171-172. It is not clear what n refers to. I expected it to indicate the sample size (number of interviews), but that does not explain small value for tucuxi. If all interviewees were asked the same question, it is not clear how the sample size varied.
L178. Is this a reference to the article co-authored by Peres or only to a personal communication in 2021? Revise for clarity.
L179ff. Note that the information described here and presented in Fig. 6 has not been listed in the interview methods. Presumably, the probability that a species will be killed, released, or escape is not usually zero or 100%. So what was the question asked – usual outcome, desired outcome, worst outcome?
Also, note that methods mentioned asking if fishers ever changed location as a result of conflict, but I did not see a result related to that question.

Discussion
L195. Are otters and crocodilians the only species ever reported to attack humans in freshwater? That seems unlikely.
L200. Is it too difficult to repeat the test using maximum size to see if your intuition is correct?
L218. Rather than simply asserting the bias, please provide some support or explanation.
L220. ‘Retaliation’ needs to be defined.
L320. Relationship of coexistence to population trends was not mentioned in Results.

Conclusions
L338. Elaborate a bit. ‘Poor documentation’ might mean only a few reports which could be due to a small species range or lack of overlap with fishing areas. Why is this more important than more investigation of widespread conflict situations or ones that result in frequent major injuries either the piscivore or people?
L344. Presumably, you need a measure of rate, i.e., a time period in addition to number of nets.

References
The references appear to be carefully compiled in general. I found only a couple of minor errors in capitalization and dash vs. hyphen, far fewer than usual. Thanks!

Figures
Some of the words in the figures are very small. To see how readable they would be, try making a reduced photocopy to the size that they would normally appear on a pdf of a PeerJ article.

Fig. 1. Nicely done. Everything necessary is included (unlike study site figures in many manuscripts I see).
Fig. 2. Although a reviewer correctly indicated that the illustration of the four species is not really needed, the article will be published online, so it is not as much of a problem. It will be useful to some readers. I suggest you leave it in, but it is your choice.
Fig. 3. The figure is misleading because years with no publications are not shown. If it is too awkward to show all the years without publications from 1977, I suggest showing number of papers per 5-year bin from 1975 – 2000. Be sure to alert readers to the bin size for this period in the caption and make it clear in the text that the correlation included years with zero publications. Please remove the horizontal lines within the panel and add a solid line for the y-axis. After 2000, provide the dates only for each 5 years (even though you provide a bar for each year) to allow you to use a larger font and make it horizontal. Increase the font size for the y-axis label.
Fig. 4. Remove the horizontal lines. Add a vertical line for the y-axis. Increase the font size for the y-axis. The y-axis label is number of articles. On Fig. 2, the label is number of studies. Is this intended to indicate a difference? If so, it should be defined in the Methods. Otherwise, use the same term consistently for the same thing to avoid confusion. The species labels will be hard to read. Perhaps the best way to allow a larger font with full species names would be to put the species names on the y-axis and the number of studies on the x-axis.
Fig. 5. Caption is not clear. I suggest something like: ‘Percent of interviewees (n = ?) who indicated that each of four species of piscivorous mammals and reptiles causes problems in general, or specifically damages fishing equipment, frightens away fish, or becomes entangled in nets. The species are black caiman (C, black bars), giant otter (O, dark grey bars), boto (B, light grey bars) and tucuxi (T, white bars).’ Note that I have added letters as reminders for reader convenience; these would be placed under the appropriate bars. Again, fonts are too small. The y-axis could be labelled ‘Percent of interviewees’. If you prefer, ‘Percent of responses’, revise the caption so that the term responses is included. Note that in previous figures you capitalized all major words of the axis labels. Please be consistent.
Fig. 6. Make adjustments as requested in previous figures (font sizes, labels, etc.). I suggest revising the caption to ‘Percent of interviewees (n = ?) who indicated the potential outcomes of entanglement in fishing nets by four species of piscivorous mammals and reptiles. Outcomes are being killed by fishers (black), dying without fisher intervention (dark grey), being released by fishers (light grey) or escaping without fisher intervention (white).’
Table 1. Heading needs to be clearer. ‘Conflict species’ is ambiguous. These could be better identified as ‘species of piscivorous reptiles and mammals from freshwater tropical and sub-tropical habitats known (or reported or suspected) to have conflict with humans (or fishers)?’ The parenthetical terms reflect that I was unable to know clearly what they represent The conflict acronym in the footnote is inconsistent with the table (and the supplementary table). Specify the Red List abbreviations.
Table S1. See my extensive comments above on the problems with this table and the need for it to be improved and incorporated into the main text.
Table S2. Add sample size to the heading.

Reviewer 1 ·

Basic reporting

The English language used is clear and unambiguous.

Authors have included conflicts regarding only mammals and reptiles. Other classes have not been included. Birds, most importantly cormorants and pelicans, are often involved in conflicts. Other than this, Field context and references have been sufficiently described.

The structure of the article is very good. Raw data have been shared.

The manuscript reports results related to the study's aims.

Experimental design

The manuscript's subjects falls well within the aims and scope of the journal.

The research question is rather simple, however, authors failed to adequately explain what they mean by "freshwater piscivores". They claimed that they reviewed and study the conflict between stakeholders and freshwater piscivores. And they proceed by including mammal and reptile species. Classes such as the birds have not been included, although bird species such as cormorants and pelican are important fish predators in freshwater ecosystems. Authors need to clarify in the Introduction (and then throughout the manuscript) what they mean by the term "freshwater piscivores". Lines 42, 43, 47, 49, 64, 67 need to be revised. They should also convincingly report why they did not include birds or piscivores from other vertebrate classes.

In lines 79-80 of Materials & Methods they reported that they searched the literature by including species name. This approach did not allow for discovering all piscivores. I would suggest to perform a more general search before narrowing down at the species level.

Other than the above, research is rigorous and methods of analysis well described and replicable.

Validity of the findings

All findings, according to the study aims and as describes in Materials & Methods, have been provided, analysed and presented in a robust and sound manner.

Discussion and Conclusions support the results.

Additional comments

Lines 31 and 222: When someone see an article about piscivores, they do not expect to read about their effect on livestock. Please explain, in the Introduction and Discussion, which species predate on livestock and to what extend.

Line 108: Give the survey instrument in a supplementary file.

Lines 135-138: Give references for all stated software.

Line 158: Differences are significant, but towards which direction? Did the severity of conflict increase with threatened status or vice versa? This could have important conservation implications and should also be highlighted in the Discussion. Authors refer to this only on the Conclusions (lines 334-336), without giving any further information or explanations.

Reviewer 2 ·

Basic reporting

Dear editor, dear authors,

many thanks for giving me the opportunity to review the manuscript entitled "Human-wildlife conflicts with freshwater piscivores". Resolving human-piscivore conflict is certainly a relevant topic. The manuscript is overal well written and uses professional English. Minor inaccuracies, missing words etc. are marked in the attached PDF document. The second research question, which deals with the severity of conflict and IUCN red list status, requires more context. The literature references section should be checked for uniformity in line with the PeerJ requirements. The article structure, figures, tables etc. is professional. The authors focus on negative implications for human when dealing with the topic "human-wildlife conflict", but could do a better job in describing associated (negative) implications for the relevant wildlife species. The authors could do a better job in connecting their case study from the Amazon to their worldwide review on human piscivor conflict. Please find specific comments in the attached PDF document.

Experimental design

The research question number 2 would require some more context in the introduction. Otherwise it remains difficult to comprehend why this analysis was conducted. The authors could give some more context why the various statistical analyses are conducted in the section "data analysis", l. 125 ff.

Validity of the findings

The underlying data have been provided. The benefit to literature is stated. Underlying data have been provided and seem to be sound.

Annotated reviews are not available for download in order to protect the identity of reviewers who chose to remain anonymous.

---

## Round 0.2 · Minor Revisions

The manuscript has improved considerably, but still needs quite a bit of work.

Major Concerns

1. Integration of the two studies

• The integration is clearer than before but still seems a bit lacking. You present the literature review first and justify the Amazonian survey as ‘providing additional evidence’ to the review. It is not clear what substantial contribution a single short-term study could add to the larger review that would be sufficient to make all that effort worthwhile, and I did not see any development of this theme in the manuscript. Is it really the case that you carried out a literature review and then decided that a survey would be helpful? If so, why was the literature review focussed only crocodilians, otters and cetaceans? It seems more likely that after the survey was done, you carried out a literature review and then decided to make it as complete and quantitative as possible. You don’t need a long explanation involving personal issues involved in developing the manuscript, but a brief, clear, logical, and honest explanation of the link between the studies would be helpful for readers to see the bigger picture.

• For clarity to readers, the presentation should keep a consistent order throughout, unless there is an extremely good reason to do otherwise. You present the literature review first and the Amazon survey second throughout the paper and then suddenly reverse the order in the Discussion.

• There are very few points where the relationship between the studies is developed in the Discussion. This is critical to expose what you learned (and what you didn’t) by joining the two studies.

2. Introduction

• The Introduction is still incomplete.

• There is no review of the relevant literature for the Amazon survey and no identification of the knowledge gaps that the study is intended to address (for both parts).

• The justification of the choice of species for the literature review is weak (‘large bodied and known to generate conflict’). If there really are no other large taxa that generate conflict, I think that more citations are required to establish this.

3. Methods

• The Methods are clearer now but still need some work. In particular, another researcher should be able to repeat them and come up with the same result.

• In the text, you refer to severity before frequency. Table 1 should follow the same order.

• Definitions for severity are awkwardly phrased. Do you mean ‘. . . frequency of documented or perceived . . .’?

• It is not clear how you distinguish the categories of fish depredation or what you mean. Does this refer to competing for the same fish population or eating fish that have been already captured (e.g., in a gill net)? The quantitative indices are specified for attacks on people but not specified for fish depredation and entanglement.

• The temporal and spatial scales of the measures are not specified. Is this the total number of cases counted over all years and all locations and all publications in the primary literature or is it something else?

• You did not respond at all to my suggestion that checking for inter-observer consistency is important. Asking a colleague (or several) to score a representative set of papers using your criteria but not knowing your scores (not necessarily the whole list) will help you identify problems in communicating the criteria and provide more credibility to your method. You must recognize that it may also reveal problems that could necessitate re-scoring.

• It is appropriate to recognize potential divisions in author roles. Indicate by initials the person(s) who scored the literature. Similarly, indicate the person(s) who conducted the surveys. Checking recent papers in the literature will show you how this is done. I can send you one of my papers where a former student collected all the field data but I helped in the interpretation if you can’t find any others. Contact me directly ([email protected]) if you need that.

• You did not respond to the suggestion that combining categories of human attacks and competition for fish and entanglement in nets may not be the best way to accomplish the goals of the review. I still believe this to be a valid comment.

• In Table 1, it should not be necessary to repeat the reference for non-fatal attacks in each definition. I suggest using an asterisk for the first definition and then providing a footnote such as ‘We defined non-fatal attacks following Sideleau and Britton (2013)’. It wasn’t clear if the reference was for both kinds of attacks or only non-fatal, so that should be clarified in the footnote

• The frequency category ‘Required CRR’ is not mentioned in the text and is unclear.

4. Results

• The Results do not explicitly connect with the objectives of the literature review.

• For reader ease and clarity, order of results should follow order of objectives. (However, preceding results pertaining to each objective, a paragraph with an overview of the number of studies and their categories etc. such as you had is totally appropriate.) You seem to be starting with Objective 3. For Objective 1, I would have expected an analysis of the types of conflict with each broad taxon and possibly with individual species in at least some cases. The percentage of studies with all taxa combined that consider attacks, competition, entanglement, etc. do not seem to meet this goal. Percent of studies does not seem to be the most relevant criterion. It needs a thoughtful interpretation.

• Objective 2 was addressed only for entanglement and attacks and only incompletely even for these.

• For Objective 3 which refers to each species, I see only broader categories.

• Don’t you need a table to show the data for Objectives 1 – 3?

• For Objectives 4 and 5, only the statistical results are presented. Readers need a table to see for themselves what the pattern is.

• Fig. 3 remains unclear with the fonts too small. Excel is not a good program for producing publishable graphs. A response in the rebuttal implying that Web of Science is not available might imply that graphical programs are not available to the first author. Nevertheless, the co-authors appear to have institutional affiliations so it is hard to imagine that none of authors have access to a graphical program either directly or through present or former colleagues.

5. Discussion

• The Discussion needs substantial revision. It is too long, overly repetitious of results, insufficiently critical of the strength of the patterns found, includes patterns that were not mentioned in the Results section and fails to discuss some patterns that were found.

• L251-261, 263-266, 293-297 are examples of sections that seem too long and too vague. Condense into conclusions or a separate section on ‘limitations of data’ or ‘research needs’ with more thought as to what is feasible and why it is needed.

• All substantial findings deserve discussion. I did not see a discussion of the positive relationship between the distance and otter conflict.

• When discussing specific aspects, you can indicate any important limitations of your survey and then devote one section of the Discussion to improvements, building on those limitations. The long sections on improvements (e.g., L251-261, 293-295) are not clearly linked to your discussion, cover multiple issues and are quite vague. To be useful, such recommendations need to be based on your experience of the literature review or your own survey or both, and they need to be practical. Think about whether and how valid data could be collected by interviews. Can fishermen really answer a question on percent of each outcome? Think about appropriate scaling; is entanglement per km of any validity without information on nets per km? Isn’t entanglement per net set more relevant to the fisher perspective and then multiplying by number of nets set more relevant to the conservation perspective?

• L279-286. Excessive repetition of Results. It is appropriate to briefly remind the reader of specific results to help them understand what you are about to discuss, but this must be tightly focussed on the relevant finding.

• L327-370. While an interesting finding, this section is much too long in comparison to the role of IUCN status in your Results. You don’t discuss the validity of the finding, perhaps excessive influence of some taxa, limitations of the severity measure, or the breadth of IUCN categories or potential political influences on IUCN status.

• I have not provided detailed comments from L370 on. I hope that the combination of the above comments and the more specific points raised below will allow you and your colleagues to make revisions to these sections similar to the preceding ones.

6. Writing

• I have made many suggestions on the pdf for improvements in the writing to make it more concise and clear.

• Many of my comments below identify wordy general comments, apologies and other material that does not belong in a tightly written scientific paper.

• My impression is that this revision is not as clearly written as the first submission. I wonder whether it was re-submitted without peer review. I spent substantial time on this revision. This should not be necessary for an editor if authors take appropriate responsibility to subject their drafts to several peers for careful reading before submission. It is critical that you solicit reading by colleagues who will be able to critically consider the content and evaluate the clarity and conciseness of the language.

Other suggestions

L4. If the corresponding author is no longer affiliated with the University of East Anglia as implied in the rebuttal, his address should include UEA as the place where the research was done as well as a ‘current address’ (for example, home address) to indicate that he is no longer there.

L109. ‘Attacks on humans were restricted to wild animals’ is not clear. Are any of the relevant species domesticated? Do you mean that attacks by animals in captivity were excluded?

L117, 127. Delete these apologies here. In the Discussion you can and should mention the specific issue when discussing the reliability of your findings, but the ‘we recognize’ statements are unnecessary and do not add anything.

L126. How can you have less than one if you are compiling accounts of conflict?

L191. If 143 is the sample size for the Spearman’s correlation, you have not done the test correctly. From the description of the result, you have examined the number of articles per year in relation to number of years, so the sample size should be 59.

L201ff. Where are the data for these patterns?

L210, 220. For chi square, you need to report d.f. as well as n.
L231. Why not more attention to this result? For example, is it a gradual increase in proportion or a sudden change? Could it be due to just one or two locations?

L204, 240, 286, 295, 306. Avoid repetitious ‘highlights that’ and use the term only when it truly highlights a point.

L238-240. I did not see this observation in your Results. Except for relationships to body mass and Red List status, I saw only summary statistics in the results. Where are the data for variation in conflict severity among species?

L240-242. This statement is not clear, and the logic is doubtful. Couldn’t a review of multiple studies just as easily reveal a difference that didn’t occur or wasn’t apparent in one particular location or small study? Isn’t it possible that one site differs from a broad trend? If you are sure that this is a valid point, make it much clearer and more decisive.

L253-254. Confusing multiple points.

L250. Why weren’t these studies mentioned in the Introduction?

L285-286. I don’t think this was in your Results.

L286-287. Frequent or infrequent? Seems contradictory and unclear.
L292. If secondary literature is defined as publications that pull together information from primary sources, there should not be any additional evidence in the secondary literature. You may be intending what is often called ‘grey literature’ (non-peer reviewed publications such as government reports, etc.) but this may not be the technical term. Make sure you are using the appropriate term for what you mean (and don’t simply take my word for it).

L292. If that would be useful, why didn’t you include it?

L308. Why does the information have to be available in one source? I would think that regional species descriptions would often give maximum size, at least for the crocs where it may vary greatly from the mean. I doubt that it is as important for otters.

L313-315. This appears to be more results from your survey that were not presented in the Results section.

L317-319. If it is not available anywhere, what is the point? If it is available, why didn’t you get it, even if it meant more work?

L320ff. You found no pattern but seem to be concluding that it is a useful measure. How can this be? The follow-up statement of ‘challenges’ is too vague to be useful to readers.

L368. Not in results; be clearer
L370ff. As noted above, I leave it to you and your co-authors to recognize similar issues of length, clarity, and completeness with the remaining text.

L453-458. Don’t repeat results in the Conclusions. I now think that implications for future studies would be clearer in its own sub-section.
References have a few minor issues.

Check the Instructions to Authors and recent PeerJ papers for proper citation of data analysis programs

---

## Round 0.3 · accepted · Accept

This manuscript has been tremendously improved. I enjoyed (re-)reading it. Thank you for taking so much care in your revision. In proofs, please give the scientific name of matrinxa on L207 where it is first mentioned instead of L331. [I worked with a couple of Brycon species in Panama during my postdoc days (1973-75!)]